# Detection of Water Content in Lettuce Canopies Based on Hyperspectral Imaging Technology under Outdoor Conditions

**Jing Zhao, Hong Li \*, Chao Chen, Yiyuan Pang and Xiaoqing Zhu**

Research Centre of Fluid Machinery Engineering and Technology, Jiangsu University, Zhenjiang 212013, China
* Correspondence: hli@ujs.edu.cn

**Abstract:** To solve the problem of non-destructive crop water content of detection under outdoor conditions, we propose a method to predict lettuce canopy water content by collecting outdoor hyperspectral images of potted lettuce plants and combining spectral analysis techniques and model training methods. Firstly, background noise was removed by correlation segmentation, proposed in this paper, whereby light intensity correction is performed on the segmented lettuce canopy images. We then chose the first derivative combined with mean centering (MC) to preprocess the raw spectral data. Hereafter, feature bands were screened by a combination of Monte Carlo uninformative variable elimination (MCUVE) and competitive adaptive reweighting sampling (CARS) to eliminate redundant information. Finally, a lettuce canopy moisture prediction model was constructed by combining partial least squares (PLS). The correlation coefficient between model predicted and measured values was used as the main model performance evaluation index, and the modeling set correlation coefficient $R_C$ was 82.71%, while the prediction set correlation coefficient $R_P$ was 84.67%. The water content of each lettuce canopy pixel was calculated by the constructed model, and the visualized lettuce water distribution map was generated by pseudo-color image processing, which finally revealed a visualization of the water content of the lettuce canopy leaves under outdoor conditions. This study extends the hyperspectral image prediction possibilities of lettuce canopy water content under outdoor conditions.

**Keywords:** hyperspectral imaging; outdoor conditions; preprocessing; feature selection; water content prediction; lettuce



## 1. Introduction

Lettuce is a typical leafy vegetable that requires a large amount of water during its growing period, and water can directly affect lettuce growth, quality, and yield [1]. Therefore, rapid and accurate lettuce water content determination is crucial for the real-time monitoring of lettuce plant growth. The hyperspectral analysis technique is widely used as an indirect analysis method for the quantitative detection of crop physiological information, and is nondestructive and rapid, thereby compensating for the shortcomings of traditional methods which destroy samples [2,3]. These advantages have led to its widespread adoption in agriculture [4–7].

Scholars across the globe have increasingly used hyperspectral imaging techniques to detect physiological crop information. Cheng et al. [8] formed a mask by thresholding the grayscale image at 672.37 nm, thereby creating a hyperspectral oilseed rape segmentation image, and a cadmium content prediction model of oilseed rape leaves was constructed using neural networks. Tung et al. [9] extracted laboratory hyperspectral images of Fengjing pak choi after water stress treatment; they preprocessed the raw spectra using smoothing and derivatives, and developed a crop water potential prediction model using modified partial least squares regression (MPLSR) with a model prediction correlation coefficient of 0.826. Sun et al. [10] extracted hyperspectral images of potato leaves using a dedicated indoor hyperspectral test frame; they screened leaf water content-sensitive

wavelengths using competitive adaptive reweighting sampling (CARS) on the raw spectral data, and constructed a water content model by partial least squares regression (PLSR) with a CARS–PLSR modeling calibration accuracy of 0.9878 and a validation accuracy of 0.9366. In Zhang et al. [11], the optimal feature subset was selected by GA–PLS from the grayscale information of lettuce leaves extracted from hyperspectral images with crop texture features, and a water content prediction model was established with a correlation coefficient *R* of 0.902 between the model predicted and measured values. In Sun Jun et al. [12], lettuce leaf hyperspectral information was collected under indoor conditions, and lettuce leaf moisture prediction models were developed by three modeling methods, MLR, PLS–ANN, and BP–ANN, respectively. Among them, the PLS–ANN model had the lowest average relative error in prediction, with 9.4515% for rosette stage and 9.1245% for nodule stage. Li Hong et al. [13] used the MCUVE–LASSO–SPA algorithm to filter characteristic wavelengths, thereby eliminating redundant information within the full spectrum, and finally selected 14 characteristic wavelengths to build a PLS lettuce canopy moisture content prediction model. The $R_C$ and RMSECV values were 0.8827 and 1.0662 for the prediction set; the $R_P$ and RMSEP were 0.9015 and 0.9287, respectively. Although the above studies used hyperspectral image technology to detect crop parameters, they had strict requirements for spectral data collection conditions [14,15], which limit the application scenarios of hyperspectral nondestructive monitoring technology. Additionally, the spectral image segmentation method was single, and failed to effectively segment the hyperspectral images of crops extracted under complex lighting conditions. The data preprocessing lacked pertinence, and the resistance to light intensity interference was poor. It was difficult to apply to practical outdoor applications. Therefore, studying the prediction method of crop canopy parameters based on hyperspectral imaging technology under outdoor conditions is of great significance.

In this paper, lettuce at the rosette stage was used as the subject of study. Outdoor lettuce canopy hyperspectral information with different water contents, as well as corresponding canopy water content, were obtained. A hyperspectral image segmentation method based on the correlation difference of spectral reflectance curves is proposed, and the canopy information in the image is segmented using this method. For complex outdoor light interference, the canopy information was extracted by using the light intensity correction method and the first derivative processing method to remove noise from the spectral data. Monte Carlo uninformative variable elimination (MCUVE), competitive adaptive reweighting sampling (CARS), and successive projections algorithm (SPA) to extract the characteristic wavelength were used. The characteristic wavelength, combined with partial least squares (PLS), produced a lettuce canopy water content prediction model, and selected the optimal model based on the model correlation coefficient and root mean square error, which was used as a hyperspectral rapid model for the lettuce canopy water content under outdoor conditions. Detection provides a reference method.

## 2. Materials and Methods

The experimental materials and treatments used in this article are described.

### 2.1. Experimental Sample

The experimental samples were selected from Romaine lettuce (*Lactuca sativa var.* 'longifolia'), which were cultivated in potted soil culture from 30 September 2021 to 3 January 2022 in the Modern Agricultural Equipment and Technology Laboratory of Jiangsu University. Three to five seeds were place in each pot, and one seedling was left pot upon reaching the five leaves one heart period (Figure 1b). Six water-holding gradients were created, and 25 samples of each gradient were irrigated with drip arrows (Type SLD109 +SLD012, Shunlv Irrigation equipment Co., Ltd, Guangzhou, China). The divisions based on the water holding gradient were [16,17]: setting the wetting layer to 15 cm; maintaining the soil water content at 40–50%, 50–60%, 60–70%, 70–80%, 80–90%, and 90–100% of the field water holding capacity; measuring the soil water content under each treatment daily

with a soil moisture sensor (Type HM-WSY, Hengmei Technology Electronics Co., Ltd, Guangzhou, China); and full irrigation until the soil water content reached the upper limit once the soil water content reached its lower limit. Lettuce was sampled uniformly as it grew to the rosette stage (Figure 1d).

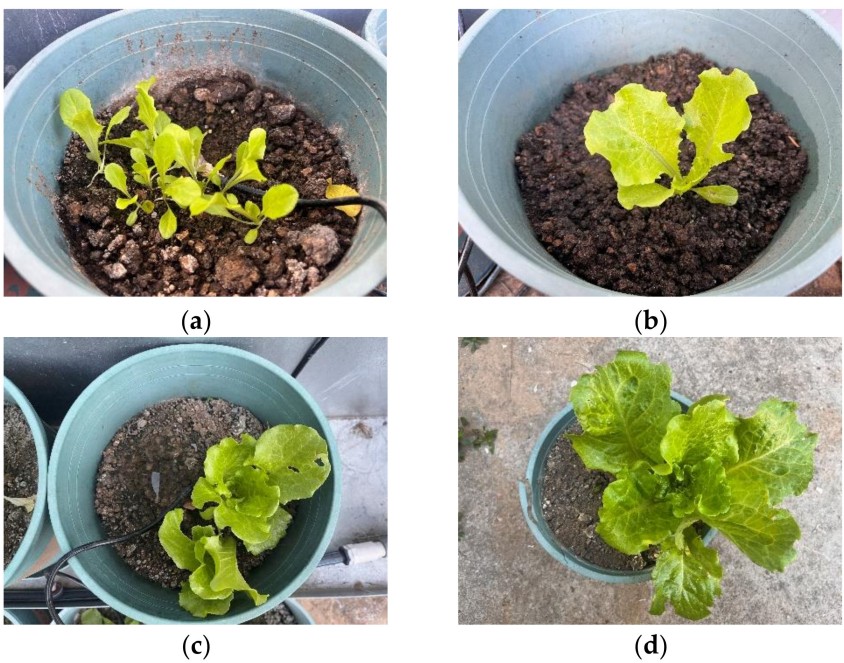

**Figure 1.** Each growth period of lettuce. (**a**) Germination period. (**b**) Five Leaves One Heart Period. (**c**) Seedling period. (**d**) Rosary period.

*2.2. Outdoor Hyperspectral Data Acquisition*

A convenient hyperspectral imaging system (Type Gaia Field, Dualix Instruments Co., Ltd., Jiangsu, China) was used to collect outdoor lettuce canopy hyperspectral information. The system consisted mainly of a hyperspectral imager (Gaia Field Pro-V10E), an imaging lens (HSIA-OLE23), a calibration whiteboard (HSIA-CT-250*280), a professional outdoor tripod (EI-740A) and data acquisition software (Spec View; Figure 2). The spectral range of the imager was 391.65~1018 nm precisely, with a spectral resolution of 2.8 nm and 176 spectral channels.

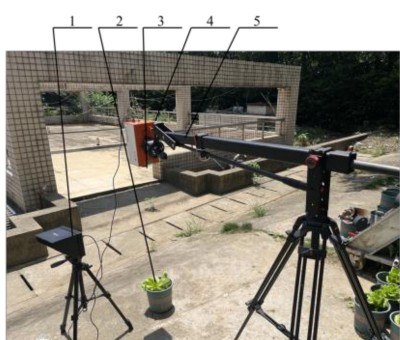

**Figure 2.** Outdoor hyperspectral imaging system. 1. Monitor. 2. Lettuce sample. 3. Sensor probe. 4. Hyperspectral imager. 5. Outdoor test stand.

To reduce the influence of weather factors on the experiment, all hyperspectral data acquisition was carried out in windless days with clear skies, and the acquisition time was consistently between 10:00–14:00 Beijing time; experimenters were also dressed in

dark clothing. For data acquisition, the hyperspectral imager sensor lens faced vertically downward, about 1 m from the vertical height of the canopy [18], and the exposure time was set to 9 ms. To avoid errors from temporal variation in outdoor light intensity, ten spectral data points were used as one sampling interval, and each interval consisted of nine lettuce canopy hyperspectral data with one standard whiteboard hyperspectral data $W$. After the sample was collected, the CCD camera lens was covered, and the hyperspectral image data point $B$ of the blackboard was acquired. To eliminate the effects of light intensity and in-camera dark current noise on spectral image quality, the spectral data at each interval were calibrated separately in black and white, and a raw image correction equation [19] was applied, namely

$$R = \frac{I - B}{W - B} \tag{1}$$

where $I$ is the raw lettuce canopy spectral data; $R$ is the corrected lettuce canopy spectral data.

Spectral images were corrected, and spectral data analyzed, using Matlab R2018b (Jack Little, Natick, MA, USA).

### 2.3. Canopy Water Content Determination

Water content was determined using the desiccation method [20], and lettuce leaf fresh mass $m_1$ was measured by removing the roots. Firstly, plants were oven dried at a constant temperature of 105 °C for 30 min, where after it was adjusted to 80 °C and plants were dried to a constant mass to determine dry mass $m_2$. The dry basis water content $w$ [21] of the canopy samples was calculated as

$$w = \frac{m_1 - m_2}{m_2} \tag{2}$$

### 2.4. Lettuce Canopy Image Extraction

The outdoor canopy layer image contains various background noise, for example ground, flower pot, and soil, and these background noises affect the lettuce canopy water content prediction accuracy. It is thus necessary to segment the lettuce canopy and background area. In Figure 3a, leaf, shaded leaf, and background pixels were selected, select each part of the area as shown in Figure 3b, and their reflectance curves are shown in Figure 3c. Since the background curve is more complex, and since there is overlap between the background curve and the lettuce sample curve in each band, the common method of setting a reflectance difference threshold to remove the background is not applicable for outdoor canopy layer image segmentation. In this paper, the correlation between wavelength reflectance curves for each pixel point was used as the basis for image segmentation. Although the blade and shadow blade reflectance differed greatly beyond 700 nm, their overall wavelength reflectance curve was highly correlated (Figure 3c).

The full-wavelength reflectance $I_C$ of pixel points in the canopy region of any lettuce plant in the hyperspectral image was selected as a reference and correlated with the full-wavelength reflectance $I_Y$ of the remaining pixel points. After calculation, the correlation coefficient matrix $X_R$ was obtained, and a correlation factor of $R = 0.93$ was used as the segmentation threshold to segment matrix $X_R$ binarization—an image is given by Figure 3d. This binarized image was used to mask the lettuce canopy layer hyperspectral image to remove background noise.

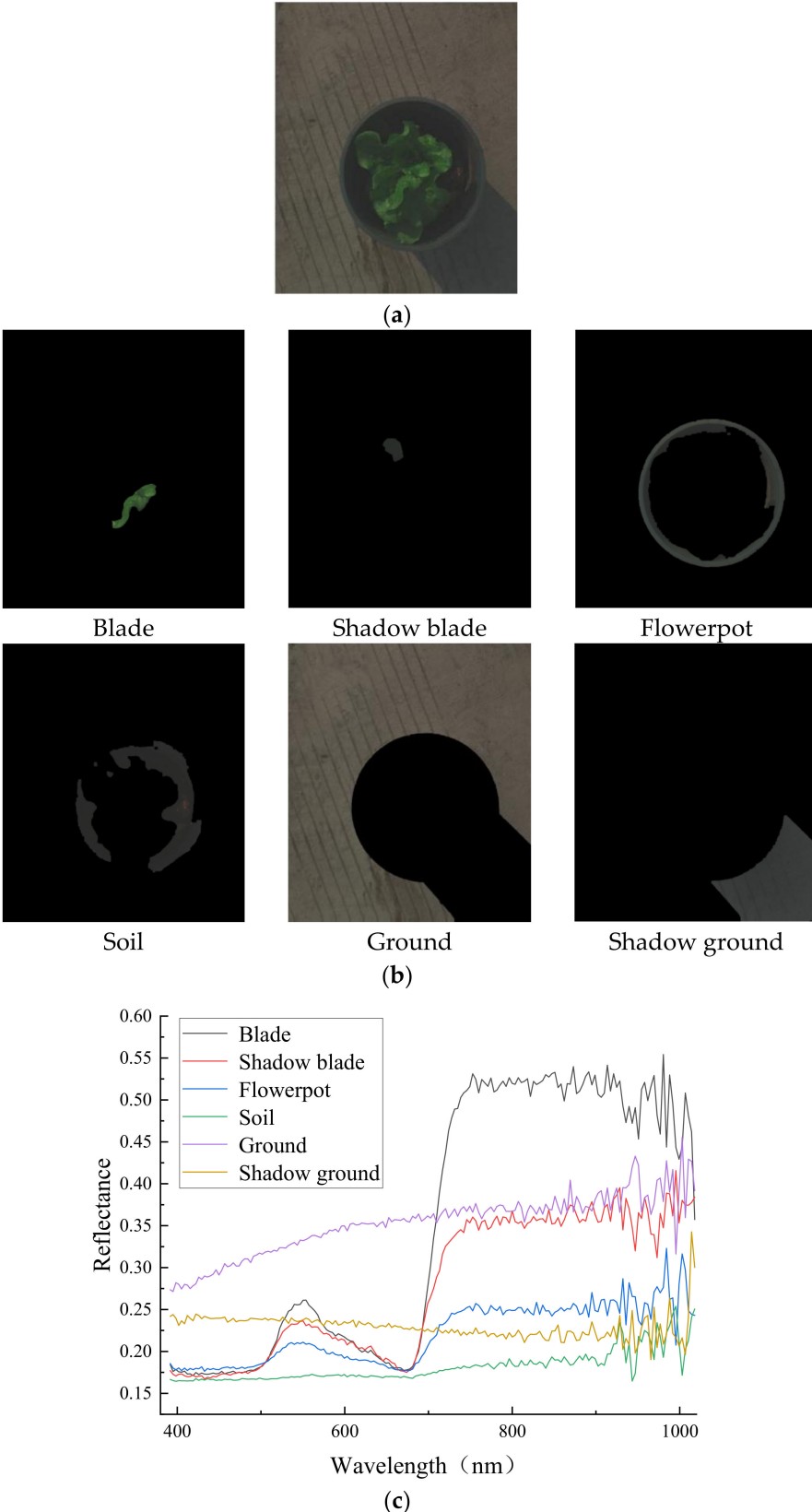

**Figure 3.** *Cont.*

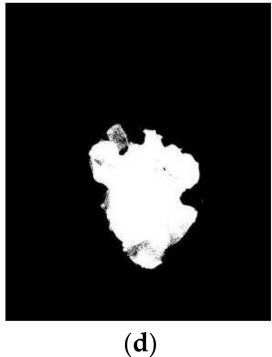

(**d**)

**Figure 3.** Lettuce canopy outdoor image processing. (**a**) Outdoor canopy images. (**b**) Selected areas for each section. (**c**) Hyperspectral reflectance of different regions. (**d**) Binarized image.

*2.5. Abnormal Sample Rejection*

In order to avoid abnormal samples generated by individual plant growth differences, instrument differences, changes in the measurement environment, and operational errors from affecting the spectral analysis results, abnormal samples need to be detected and retained or rejected according to the actual situation. In this paper, we determine whether to exclude samples based on the Mahalanobis distance between the spectral curves of each sample [22].

After converting the spectral image of lettuce into a data matrix, it will become a matrix $A$ of $n \times k$.

Calculate the average spectrum of $n$ samples:

$$\overline{A} = \sum_{i=1}^{n} A_{ij} / n \tag{3}$$

where $A_{ij}$ is sample spectral matrix element; $n$ is the number of samples; $j$ is the wavelength serial number; $\overline{A}$ is the average value of the sample spectrum.

$$A_u = A - \overline{A} \tag{4}$$

where $A_u$ represents the spectral matrix after centralization; $A$ represents the raw spectral matrix; $\overline{A}$ represents the average coordination matrix of spectrum.

The covariance array of the original standard spectral data set is then calculated.

$$M = A_u^T A_u / (n - 1) \tag{5}$$

where $M$ is the covariance array of the standard spectral data set; $A_u^T$ represents the transpose of the spectral matrix after the centralization process; $A_u$ represents the centralized spectral array; and $n$ represents the number of samples.

The Mahalanobis distance between the calibration set sample data and the average spectral data was calculated from both.

$$D^2 = \left( A_i - \overline{A} \right) M^{-1} \left( A_i - \overline{A} \right)^T \tag{6}$$

where $M^{-1}$ is the inverse matrix of $M$.

Out of a total of 143 samples, after calculating the Mahalanobis distance between the spectra of each sample and the average spectrum, the Mahalanobis distance of each sample is shown in Figure 4. Four samples have a significantly higher Mahalanobis distance than the other samples, so they are judged as abnormal samples and are eliminated, leaving 139 samples.

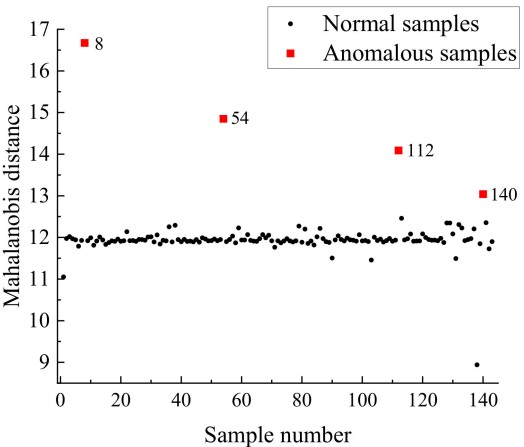

**Figure 4.** Mahalanobis distance for each sample.

### 2.6. Spectral Preprocessing

A large amount of stray light under outdoor conditions and baseline drift, caused by changes in sunlight angle, can be included in the raw outdoor hyperspectral data (Figure 5), and these interfering factors can affect model accuracy. Therefore, spectral preprocessing is required before building a multivariate calibration model [23]. In this paper, three preprocessing methods, namely the Savitzky–Golay convolutional smoothing (S–G), standard normal variables transformation (SNV), and 1st derivative were used to preprocess the raw hyperspectral data, compare the modeling effects with each other, and select the data with the best modeling effect for subsequent analyses.

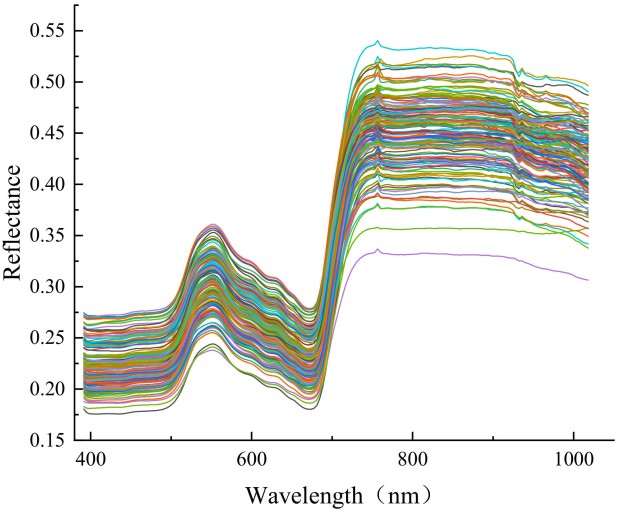

**Figure 5.** Outdoor hyperspectral raw data.

Savitzky–Golay convolutional smoothing (S–G) [24] is applied to eliminate irregular random noise and emphasizes the fundamental role of the centroid. The S–G convolution smoothing value of the wavelength point is:

$$X_{k,smooth} = \overline{x_k} = \frac{1}{H}\sum_{i=-w}^{+w} x_{k+i} h_i \tag{7}$$

The formula for calculating $H$ is:

$$H = \sum_{i=-w}^{+w} h_i \tag{8}$$

where $H$ is the normalization factor; and $h_i$ is the smoothing factor. Multiplying the measured values by the smoothing factor $h_i$ minimizes the effect of smoothing on useful information; S–G convolutional smoothing hyperspectral data are shown in Figure 6.

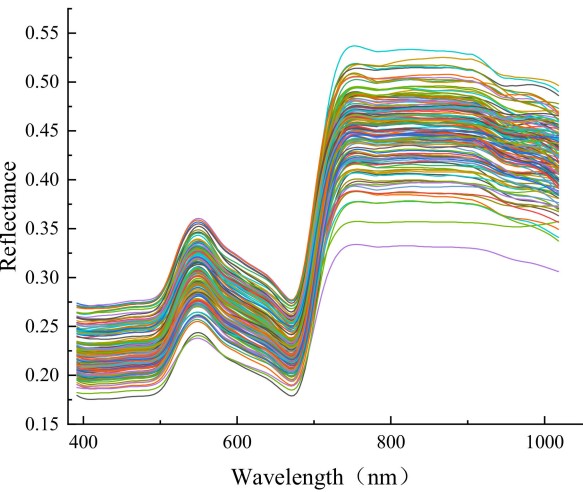

**Figure 6.** Savitzky-Golay convolutional smooth data.

The standard normal variables transformation (SNV) [25] is mainly used to eliminate the effects of sample particle size, surface scattered light, and optical range shift on the spectrum. The calculation formula is:

$$x_{SNV} = \frac{x - \overline{x}}{\sqrt{\frac{\sum_{k=1}^{m} (x_k - \overline{x})^2}{m-1}}} \tag{9}$$

$$\overline{x} = \left(\sum_{k=1}^{m} x_k\right) / m \tag{10}$$

where $m$ is the number of wavelength points; $k = 1, 2, \cdots, m$. The data after SNV are shown by Figure 7.

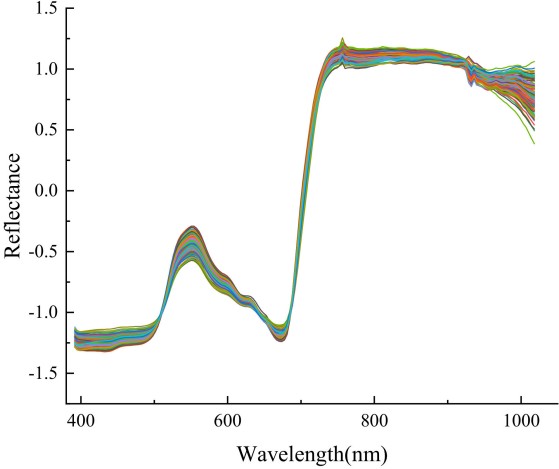

**Figure 7.** SNV data.

The main role of the derivative [26] is to eliminate baseline and other background interference, and it is a common method used for diffuse reflectance spectra. The 1st derivative of the differential width $g$ of the spectrum $X_K$ at wavelength $k$ is:

$$x_{k,1st} = \frac{x_{k+g} - x_{k-g}}{g} \tag{11}$$

The spectral data after the 1st derivative processing are shown by Figure 8.

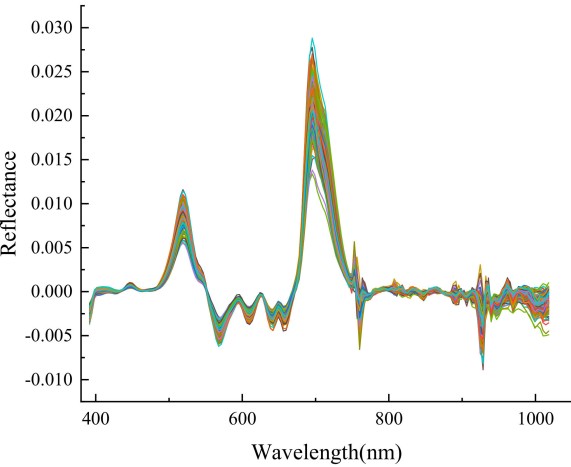

**Figure 8.** 1st derivative data.

### 2.7. Characteristic Wavelength Screening

After preprocessing and optimizing the outdoor hyperspectral data, problems remain, such as a large number of data variables and overlapping spectral information. Such redundant data must be removed to reduce the number of data dimensions and to simplify algorithm complexity. In this paper, we used Monte Carlo uninformative variable elimination (MCUVE) to eliminate irrelevant variables and combined competitive adaptive reweighting sampling (CARS) and successive projections algorithm (SPA) to screen wavelengths.

#### 2.7.1. Monte Carlo Uninformative Variable Elimination

The Monte Carlo uninformative variable elimination method [27] is based on the Monte Carlo multiple resampling modeling. The regression coefficient stability value measures wavelength importance by making full use of the internal correlation between samples, and wavelengths with a stability below the specified threshold value are screened out by setting a threshold.

#### 2.7.2. Competitive Adaptive Reweighting Sampling

The competitive adaptive reweighting sampling (CARS) [28] uses MCUVE or random sampling to select a portion of the calibration set samples for PLS modeling. The adaptive reweighted sampling method is used for screening, and wavelengths with large absolute PLS regression coefficients are retained to obtain a series of wavelength variable subsets. Cross-validation is then used to model each wavelength variable subsets, and the optimal wavelength variable subset is selected by the model minimum root mean square error of cross-validation (RMSECV).

#### 2.7.3. Successive Projections Algorithm

The successive projections algorithm (SPA) [29] is a forward variable cyclic selection method. It uses a vector projection analysis to find the set of variables containing the minimum redundant information required to minimize the covariance between vectors, reduce model complexity, and improve modeling speed and efficiency.

### 3. Results and Analysis

The results obtained by each method are described in tables and pictures, and the obtained results are analyzed to determine the final pre-processing method and feature selection method based on the merits of the results.

#### 3.1. Lettuce Canopy Dry Base Water Content Statistics

Due to errors in the sampling of the spectral data, four abnormal samples were excluded, and the remaining 139 samples represented the total sample. Sample dry basis moisture content ranged between 9.6353 and 16.500, with a mean of 12.5583 and a standard deviation of 1.3386. The samples were divided in a 3:1 ratio into modeling and prediction sets according to the content gradient method [30]; 109 samples were thus used for the modeling set and the remaining 30 samples were used for the prediction set (Table 1). This division method ensures that prediction water content samples are within the modeling set sample water content range, and the sample water content is therefore more uniformly distributed.

**Table 1.** Dry basis moisture content of sample and results of sample sets partition.

| Dataset | Sample Size | Maximum Value | Minimum Value | Average Value | Standard Deviation |
|---|---|---|---|---|---|
| Total Sample | 139 | 16.5000 | 9.6353 | 12.5583 | 1.3386 |
| Modeling set | 109 | 15.5106 | 9.6774 | 12.4624 | 1.2551 |
| Prediction set | 30 | 16.5000 | 9.6353 | 12.9348 | 1.5778 |

#### 3.2. Lettuce Canopy Image Extraction Accuracy

Evaluation of the segmentation effect of reflectance threshold method and correlation difference method by AOM and ME.

##### 3.2.1. Segmentation Accuracy

To evaluate lettuce canopy segmentation accuracy, the original lettuce images were manually segmented and compared using the reflectance thresholding method and the correlation difference segmentation results as described in Section 2.4, respectively; the area overlap measure (AOM) [31] and misclassified error (ME) [32] were used as evaluation indices to measure segmentation performance. AOM is specifically used to analyze the deviation between the resultant area of the segmentation algorithm and the manual segmentation area. Larger AOM values indicate better segmentation effects, with an AOM value of 1, representing the best segmentation effect. ME represents the ratio of the number of mis-segmented pixels to the total number of manually segmented pixels, where the number of mis-segmented pixels is the sum of the under and over-segmented regions. Smaller ME values indicates better segmentation effects, with an ME value of 0 representing the best segmentation effect.

##### 3.2.2. Analysis of Segmentation Results

Five of the 109 modeling set hyperspectral images were randomly selected as samples for segmentation accuracy analysis, as shown in Figure 9a, while Figure 9b shows the segmentation results using reflectance thresholding, and Figure 9c shows the segmentation results using correlation differences. By comparing Figure 9a with Figure 9b, it is clear that the correlation difference segmentation leads to a higher accuracy, fewer mis-segmented pixels, and better performance in segmenting the shaded leaf parts, as well as clearer pixel segmentation at the canopy edges.

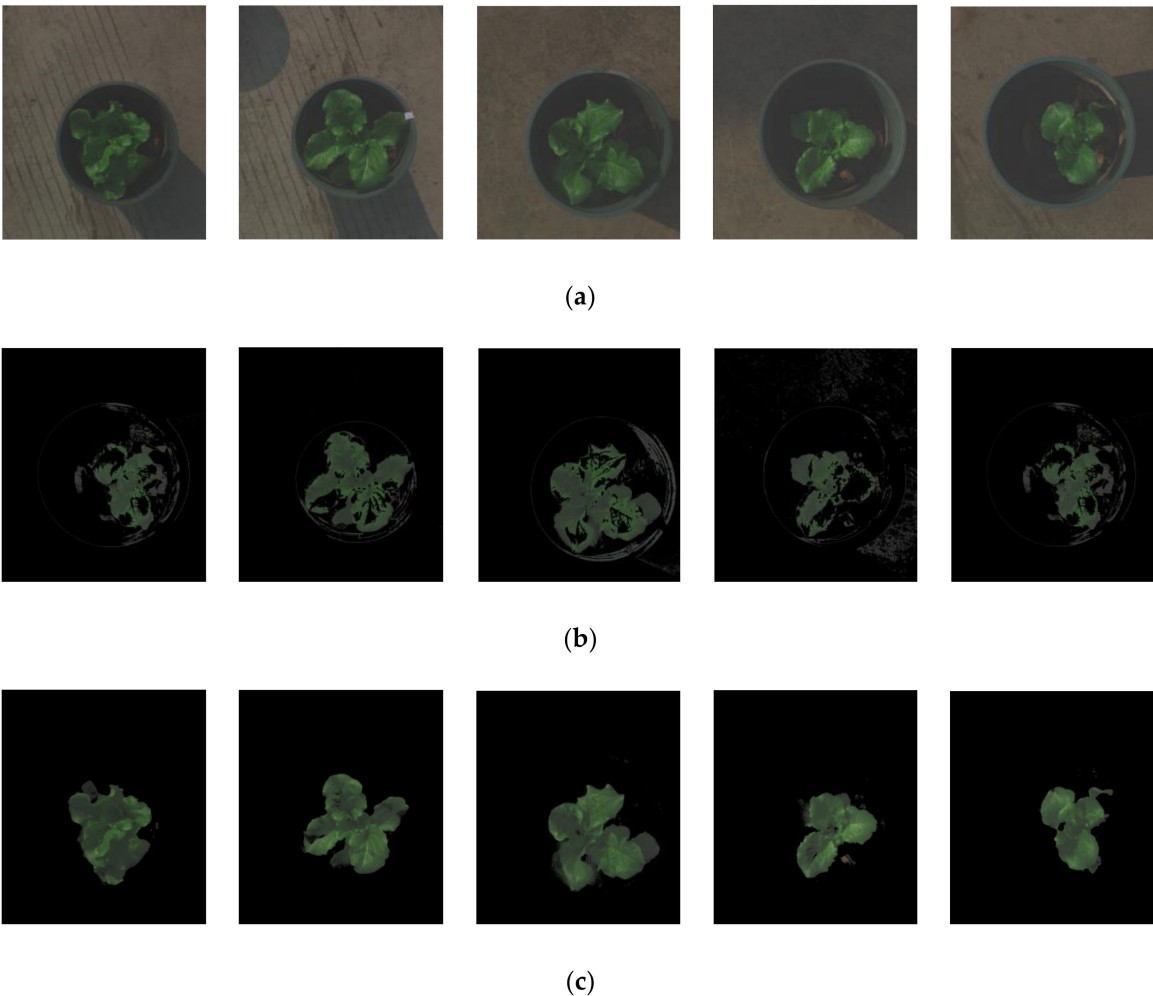

**Figure 9.** Comparison diagram of sample segmentation results. (**a**) Sample raw RGB image. (**b**) Segmentation results of reflectance threshold segmentation method. (**c**) Correlation difference method segmentation results.

After segmentation using the reflectance threshold method, sample AOM values differed greatly, with a maximum AOM value of 0.7095, a minimum AOM value of 0.4192, and a variance of 0.1235 (Table 2). The sample AOM data show that the reflectance threshold segmentation method is influenced by disturbance factors such as lettuce canopy growth and outdoor light angle during segmentation, which causes large fluctuations in the segmentation results. The AOM mean sampling value was low at 0.5640. After segmentation using the correlation segmentation, the AOM mean sampling value was 0.9252, with a variance of 0.0275, and the AOM mean value was high and the segmentation results stable for each sample. The mean ME segmentation value for the reflectance threshold method was 0.2326, with a variance of 0.0869, and the mean ME segmentation value for the correlation segmentation was 0.0292 with a variance of 0.0143. After comparing the data, the accuracy and stability of the correlation segmentation results improved compared to the reflectance threshold method, indicating that the correlation segmentation is more effective for segmenting lettuce canopy leaves under complex outdoor lighting conditions.

**Table 2.** Image segmentation performance evaluation.

| Sample | Area Overlap Measure (AOM) | | Misclassified Error (ME) | |
|---|---|---|---|---|
| | Reflectance Threshold | Correlation Difference | Reflectance Threshold | Correlation Difference |
| 1 | 0.6726 | 0.9571 | 0.2036 | 0.0109 |
| 2 | 0.7095 | 0.9457 | 0.1021 | 0.0308 |
| 3 | 0.5326 | 0.9047 | 0.2453 | 0.0189 |
| 4 | 0.4192 | 0.8911 | 0.3328 | 0.0441 |
| 5 | 0.4859 | 0.9274 | 0.2791 | 0.0414 |
| Average | 0.5640 | 0.9252 | 0.2326 | 0.0292 |
| Variance | 0.1235 | 0.0275 | 0.0869 | 0.0143 |

*3.3. Light Intensity Correction*

Under outdoor natural lighting conditions, due to uneven lighting and different lettuce leaf angles, some overexposed and shadow leaves are produced, which affect prediction accuracy. In this paper, the light intensity correction method [33] was used to study the segmented lettuce canopy region, and the average pixel point reflectance $I_m$ in each waveband was calculated as:

$$I_m(\lambda) = \frac{1}{N}\sum_{i=1}^{N} I(\lambda, i) \tag{12}$$

in the formula, $N$—Total pixel points within the lettuce canopy after splitting. $I(\lambda, i)$—Reflectance at wavelength $\lambda$ for the $i$-th pixel point in the lettuce canopy region.

The pixel points are spectrally normalized to ensure that the data are in the same range, where after the spectral reflectance of each pixel point is corrected, and calculated as:

$$I_C(x, y, \lambda) = \frac{I(x, y, \lambda)}{\max_\lambda(I(x, y, \lambda))}\max(I_m(\lambda)) \tag{13}$$

in the formula, $I(x, y, \lambda) - (x, y)$—Reflectance of pixel dots at wavelength $\lambda$ $I_C(x, y, \lambda)$—Spectral reflectance after light intensity correction.

Figure 10b shows the RGB image of the lettuce hyperspectral curve after correction, comparison with Figure 10a shows that the brighter part has reduced brightness, while darker part has an increased brightness, the uneven light intensity problem of the lettuce canopy image was therefore improved by this method.

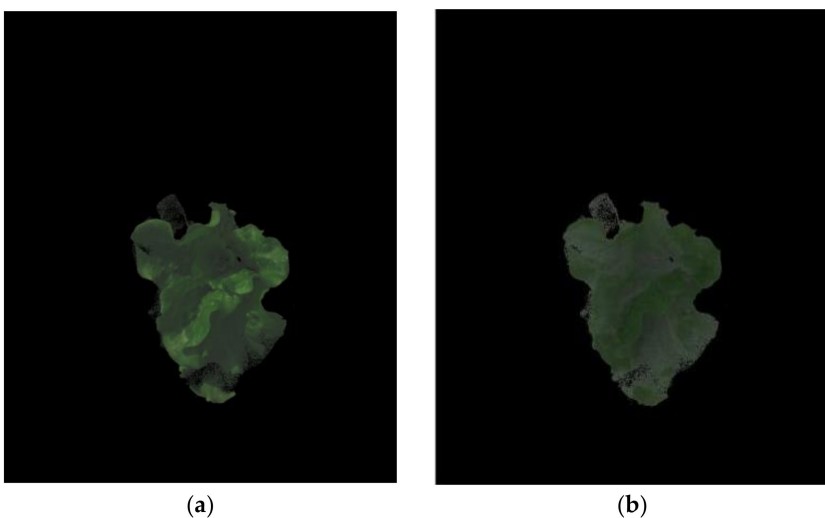

(a)          (b)

**Figure 10.** Light intensity correction comparison image. (**a**) Canopy images. (**b**) Light intensity corrected canopy images.

### 3.4. Performance Comparison of Different Pretreatment Methods

The lettuce canopy spectral data were processed by Savitzky–Golay convolutional smoothing (S–G), standard normal variables transformation (SNV), and 1st derivative methods. The preprocessed data and lettuce canopy water content were used as input values, and the water content prediction model was established using PLS. Furthermore, the modeling set correlation coefficient ($R_C$), root mean square error of cross-validation (RMSECV), prediction set correlation coefficient ($R_P$), and root mean square error of prediction set (RMSEP) of the modeling set were all used as model evaluation criteria (see Table 3 for results).

**Table 3.** Modeling effects of each pretreatment method.

| Preprocessing Methods | Master Score | Modeling Set | | Prediction Set | |
|---|---|---|---|---|---|
| | | $R_C$ | RMSECV | $R_P$ | RMSEP |
| Raw data | 11 | 76.28% | 0.8012 | 79.34% | 0.9677 |
| S-G | 17 | 81.14% | 0.7327 | 77.38% | 1.0101 |
| 1st derivative | 11 | 82.21% | 0.7213 | 81.25% | 0.9103 |
| SNV | 12 | 80.14% | 0.7484 | 75.41% | 1.0262 |

Compared with the untreated data, the correlation coefficient and cross-validation error of the S–G smoothed and SNV-treated data models were optimized (Table 3). In contrast, the correlation coefficient of the prediction set decreased and the root mean square error increased, while the modeling set $R_C$, RMSECV, prediction set, $R_P$, and RMSEP were all optimized after the 1st derivative preprocessing. The 1st derivative preprocessing method is therefore more advantageous for outdoor hyperspectral canopy data.

The 1st derivative processed spectral data were mean centered (Figure 11) after processing the spectral matrix column mean to zero. By correlating spectral variation with the variation of the measured attribute, the spectral variation information can be improved, which simplifies the regression model calculation during the following step.

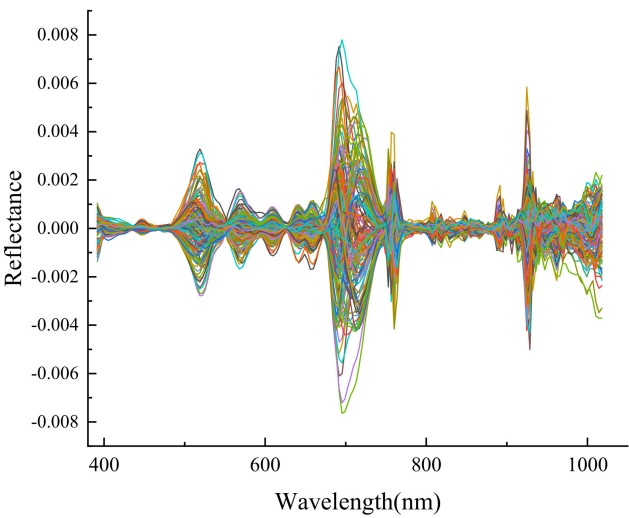

**Figure 11.** 1st derivative-mean centered dispose.

### 3.5. Dimensionality Reduction of Canopy Hyperspectral Data

By dimensionality reduction of hyperspectral data, the complexity of the prediction model can be reduced and the accuracy of the prediction model can be improved.

### 3.5.1. Eliminating Irrelevant Variables Based on MCUVE

Superfluous information was removed from the 176 spectral variables using MCUVE, which is based on 1000 Monte Carlo samples, and 109 samples were randomly selected from 139 samples each time to build the PLS model. The stability indices of the correlation coefficients corresponding to each wavelength variable are shown in Figure 12a. The absolute values of the variable stability indices were sorted from largest to smallest, and 109 samples were used to build the PLS forward additive model, while 30 samples were used as the prediction set. RMSEP prediction set changes are shown in Figure 12b: the lowest prediction set root mean square error of 0.9812 was obtained when the number of variables was 50, and this set of 50 variables was used as the characteristic variables.

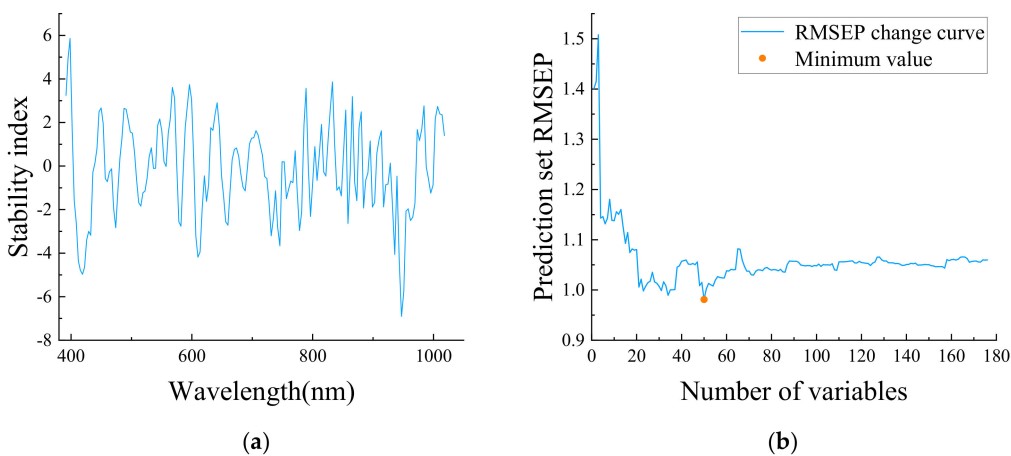

**Figure 12.** MCUVE wavelength screening process. (**a**) Stability of correlation coefficient for each wavelength. (**b**) Variation with number of variables prediction set RMSEP.

### 3.5.2. Feature Wavelength Extraction Based on MCUVE–CARS

The competitive adaptive reweighting sampling (CARS) method was used to screen the 50 spectral variables after MCUVE selection, whereafter 28 spectral variables were obtained. The number of CARS samples was set to 200, and a 10-fold cross-validation method was used (see Figure 13a for the variable selection process). From Figure 13b, a minimum RMSECV of 0.7353 was obtained when the number of iterations was 41, and 28 variables were finally selected as the wavelength variables (395.00, 398.40, 412.10, 415.50, 418.90, 425.80, 432.60, 446.30, 449.80, 487.70, 592.40, 596.00, 599.50, 606.50, 610.10, 638.40, 641.90, 742.10, 745.70, 778.20, 789.10, 829.10, 854.80, 858.40, 880.50, 935.90, 947.00, and 984.30 nm).

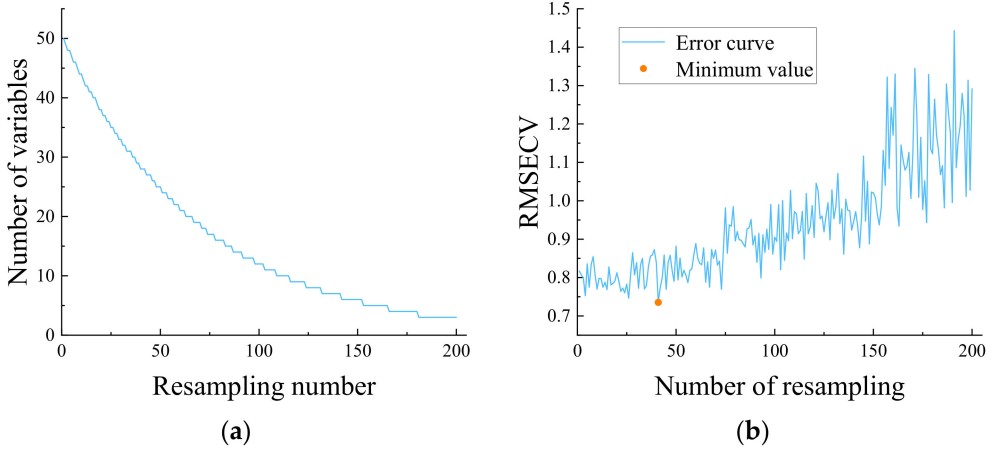

**Figure 13.** MCUVE–CARS screening process. (**a**) Variable selection process. (**b**) RMSECV trends.

### 3.5.3. Feature Wavelength Extraction Based on MCUVE–SPA and MCUVE–CARS–SPA

The SPA was used to further filter the 50 spectral variables after MCUVE selection to eliminate overlapping bands of information present in the remaining variables. The maximum number of selected variables was set to 20, and Figure 14a shows that the root mean square error gradually decreased with an increasing number of added bands until 16 variables were selected. The smallest root mean square error (RMSEP:0.8837) was finally obtained when the 16th variable was added, after which the error value slowly increased. The final 16 wavelength variables were selected from 50 variables (Figure 14b); these were: 398.40, 415.50, 429.20, 432.60, 487.70, 596.00, 656.10, 778.20, 789.10, 832.80, 854.80, 858.40, 865.80, 935.90, 943.30, and 1014.30 nm.

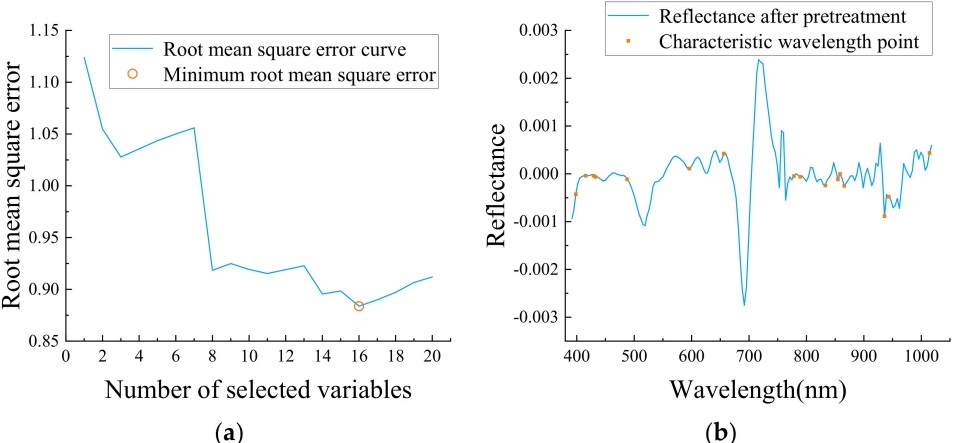

**Figure 14.** MCUVE–SPA screening the optimal combination wavelength results. (**a**) Trend of RMSEP value with the number of wavelengths. (**b**) Corresponding characteristic wavelength.

Based on the density of the variable distribution in the latter part of the MCUVE–CARS algorithm screening, some covariance existed, and the SPA algorithm was used to compress the wavelength variables again for model simplification. The RMSEP variation trend was similar to the MCUVE–SPA algorithm in Figure 15a, and the number of variables screened out was similar. The variables were added to 13 to obtain the minimum root mean square error (RMSEP: 0.8717). The characteristic wavelengths screened (Figure 15b) were: 398.40, 415.50, 432.60, 449.80, 596.00, 610.10, 745.70, 829.10, 854.80, 858.40, 935.90, 947.00, and 984.30 nm.

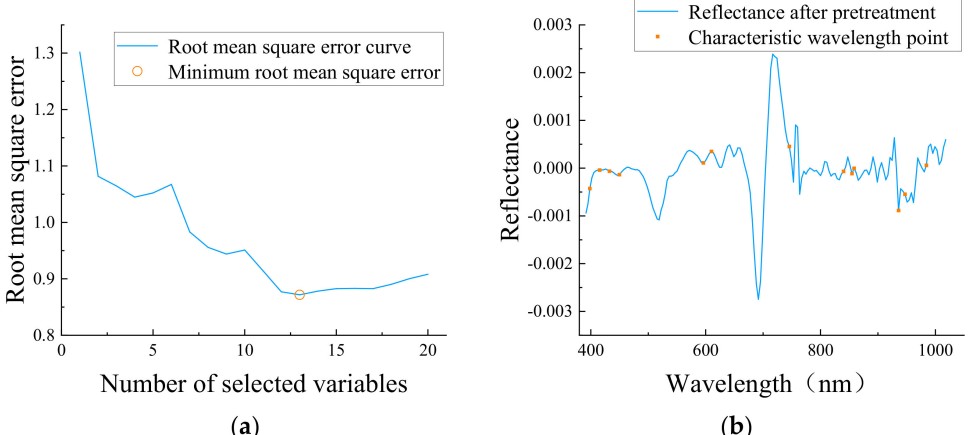

**Figure 15.** MCUVE–CARS–SPA screening the optimal combination wavelength results. (**a**) Trend of RMSEP value with the number of wavelengths. (**b**) Corresponding characteristic wavelength.

### 3.6. Modeling Results and Analysis

Lettuce canopy spectral data and water content were used as input values, and PLS was used to model the feature variables selected by different methods. The optimal principal component fraction was selected separately to build a water content prediction model. The modeling set correlation coefficient ($R_C$), root mean square error of cross-validation (RMSECV), prediction set correlation coefficient ($R_P$), and root mean square error of prediction set (RMSEP) were all used as evaluation criteria (see Table 4 for results).

**Table 4.** PLS modeling results based on different combinations of wavelength selection methods.

| Models | Number of Variables | Modeling Set | | Prediction Set | |
|---|---|---|---|---|---|
| | | $R_C$ | RMSECV | $R_P$ | RMSEP |
| PLS | 176 | 82.32% | 0.7109 | 82.38% | 0.8911 |
| MCUVE–PLS | 50 | 82.21% | 0.7142 | 82.68% | 0.9153 |
| MCUVE–CARS–PLS | 28 | 82.71% | 0.7049 | 84.29% | 0.8629 |
| MCUVE–SPA–PLS | 16 | 79.91% | 0.7530 | 83.52% | 0.8900 |
| MCUVE–CARS–SPA–PLS | 13 | 78.86% | 0.7700 | 84.67% | 0.8487 |

The PLS model constructed from the full-spectrum data after 1st derivative-mean centering had relatively good predictive power (Table 4), but the model contained 176 band variables. An increased number of variables have a greater impact on the subsequent calculation, as well as calculation stability. After MCUVE screening to remove redundant information in the full-spectrum wavelength band, the predictive power of the established MCUVE–PLS model was consistent with that of the full-band PLS prediction model, with the modeling set correlation coefficient ($R_C$) reduced by 0.1215%, corresponding to an increase in RMSECV by 0.0033; while the prediction set correlation coefficient ($R_P$) increased by 0.3642%, and RMSEP increased by 0.0242, the wavelength variables were reduced to 50. The prediction results of the MCUVE–CARS–PLS model were better in all aspects compared to both the full-band PLS and MCUVE–PLS models, and the modeling set and prediction set correlation coefficients improved to 82.71% and 84.29%, respectively, while the modeling and prediction errors were reduced to 0.7049 and 0.8629, respectively; the number of wavelength variables was reduced to 28, while the prediction ability was enhanced. The MCUVE–SPA–PLS and MCUVE–CARS–SPA models contained similar numbers of spectral variables, namely 16 and 13, respectively. The number of spectral variables involved in modeling was significantly reduced, and the prediction performance of the modeling set decreased significantly, while the predicting power of the prediction set significantly improved. Because SPA is an unsupervised variable selection method, the correlation between the selected independent variables is low and a loss of variable details occurs. This leads to a lowered modeling set predictive performance; however, the lower number of variables reduces the overfitting phenomenon and also improves model stability, thereby improving the prediction set. By comparing the four models, it is evident that the MCUVE method can better eliminate redundant variable information. However, redundant information is still present and model accuracy does not significantly improve. The CARS method has a strong screening ability, and improves model performance to a larger extent, while the SPA compression effect is the strongest, but it suffers from insufficient useful information for modeling. Overall, although the number of modeling variables for MCUVE–CARS–PLS was not the smallest among the four models, this model ensured the greatest prediction accuracy for the modeling and prediction sets, while still reducing the number of variables significantly, and had the best comprehensive performance and application capability. Figure 16 shows the prediction results of the modeling and prediction sets of the MCUVE–CARS–PLS model.

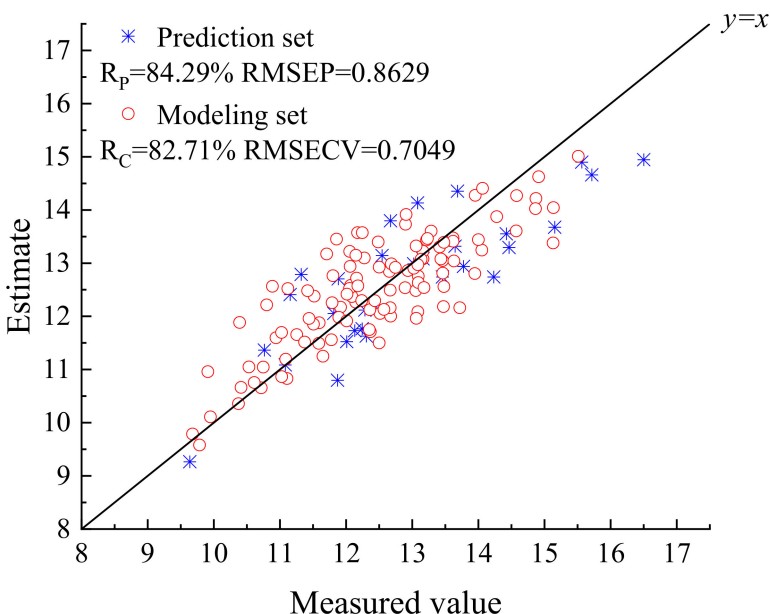

**Figure 16.** MCUVE–CARS–PLS model modeling set and prediction set results.

### *3.7. Visualization of Water Content Distribution of Lettuce Canopy Dry Base*

The MCUVE–CARS–PLS model was used to estimate the water content of each pixel point in the lettuce canopy region, and the pseudo-color image technique was applied to generate a water content distribution map, with different colors and shades representing different water contents, and white areas representing the background (Figure 17). In Figure 17a, the lettuce canopy, with a dry basis water content of 9.3352 was mainly dark blue and blue-green, while the dry basis water content was concentrated in the range of 5–15, with a mean value of 9.9256. This differed by 0.5904 from the actual mean value and was smaller than the model error mean value. In Figure 17b, the lettuce canopy, with a dry basis water content of 15.15385, was mainly green, and the water content ranged between 10–25, with a mean value of 15.7662. This differed from the actual water content by 0.61235. The water content in Figure 17b is higher than in Figure 17a, which is consistent with the actual measured water content. Whole plant water content distribution can be more directly visualized by using a visual distribution map of lettuce canopy water content.

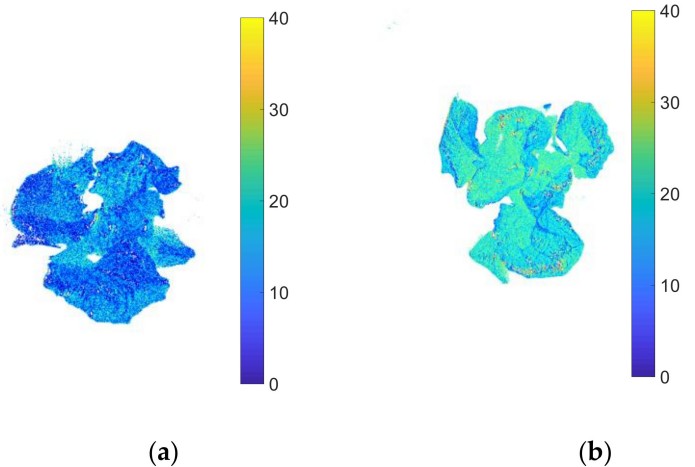

(**a**)　　　　　　　　　　　　　　　(**b**)

**Figure 17.** Moisture content distribution of lettuce canopy dry base. (**a**) Dry base water content 9.3352. (**b**) Dry base water content 15.15385.

To verify the reliability of the visualization prediction results, 10 healthy leaves were selected, which were located at different lettuce canopy positions, and their mean leaf water content values were compared with the predicted mean values of each pixel point in the respective leaf area (Table 5). As shown in Table 5, the mean predicted values of leaf water content and leaf pixel point water content were highly similar, while the root mean square error of prediction was 0.7248, which was smaller than RMSEP, thus indicating reliable visualization results.

**Table 5.** Mean value and comparison results of dry basis moisture in different lettuce leaf parts.

| Sample | Average Water Content | Predicted Mean Value | Prediction Root Mean Square Error |
|---|---|---|---|
| 1 | 12.5119 | 11.8726 | |
| 2 | 13.0226 | 13.5659 | |
| 3 | 13.4793 | 14.1203 | |
| 4 | 10.7640 | 11.5225 | |
| 5 | 11.5929 | 10.7398 | 0.7248 |
| 6 | 10.4118 | 11.0234 | |
| 7 | 15.5106 | 14.8997 | |
| 8 | 14.4628 | 15.2131 | |
| 9 | 12.1731 | 11.2341 | |
| 10 | 12.0544 | 12.8563 | |

## 4. Discussion

In this paper, the 1st derivative was used to preprocess the raw data by MCUVE–CARS to select the characteristic wavelengths and establish a PLS moisture prediction model with a model prediction set correlation coefficient $R_P$ of 0.8492. $R_P$ of 0.8492. The research effects and shortcomings of this paper are discussed by referring to the research results of peers.

### 4.1. Comparison of Predicted Effects

Reference [13] developed a lettuce moisture PLS prediction model for hyperspectral samples collected under laboratory ideals with an $R_P$ of 0.9015 for the model prediction set after screening spectral feature wavelengths by MCUVE–LASSO–SPA. In this paper and Reference [13], both of which target hyperspectral detection of lettuce samples for water content, similar prediction accuracy was obtained by adding targeted pretreatment methods with a decrease in correlation coefficient of 0.0586 after modifying the collection environment to an outdoor area with more interfering conditions in this paper. Although the accuracy was slightly reduced, the environmental restrictions on the use of spectral detection techniques were reduced.

In reference [34], hyperspectral data of outdoor potted wheat leaves were collected and combined with RVI and NDVI to build a wheat LWC model with the best modeling set $R^2 = 0.889$ and prediction set $R^2 = 0.891$. Also, reference [34] extracted crop spectral samples outdoors, but only for individual target points of the leaves, and the extracted leaf area was limited to the central region of the leaves. Although the prediction accuracy of literature 33 is higher than that of this paper, the single point extraction of hyperspectral data cannot image the whole canopy area, resulting in the inability to make guidance for the water distribution of the whole plant. The simple extraction area and small area will naturally reduce the error and get better model results.

Nondestructive moisture content testing of field crops can also be achieved by using UAVs with hyperspectral equipment. The authors of reference [35] used UAV spectroscopy to predict the water content of maize canopies in large fields. The correlation coefficient R for the prediction of crop water content of the overall field by the drone spectra was 0.93. Compared with this paper, reference [35] requires model corrections in combination with other crop growth indices in addition to the water content of the crop itself, collects a wider variety of data, and can only predict the overall water content of the maize canopy in the field, with no predictive capability for individual crops.

Of course, nondestructive testing is not only achieved by one method of hyperspectral testing. The authors of reference [36] use the relationship between crop electrical properties and crop water content to establish a prediction model, and the model predicts a coefficient of determination R2 up to 0.9154, but this method has pressure acting on the crop surface during testing, and cannot achieve completely nondestructive testing. The experimental accuracy is also affected by the crop type, temperature, and pressure of the test platen, which is significantly lacking compared with the hyperspectral nondestructive testing method.

In summary, the research results of this paper have practical significance in outdoor hyperspectral detection.

### 4.2. Improvement Methods

Since this experiment was conducted outdoors, the sampling time span is large and the light intensity will change with time, which will have an impact on the accuracy of the model. In references [37,38], after the study, it was found that as the light intensity changes, the spectral reflectance curve of the crop will also change, and this error is difficult to eliminate through correction and pre-processing. Subsequent studies will consider the effect of light intensity changes on the model.

In the reference [39], good segmentation results were achieved by logarithmic pre-processing and feature selection of the waveband before image segmentation. Image segmentation in the subsequent research can add image preprocessing and feature band selection to further improve the image segmentation accuracy.

Reference [40], the problem of burr in the front and back part of the spectral curve, which is caused by the detection limitation of the spectral camera, is mentioned in the study. To avoid the influence of the noise band on the model, the noise interval was removed by a VNIR second-order spectral standard deviation threshold. At present, we also found this problem with a more obvious noise interval at the tail end of the spectrum, which can be solved by referring to the method in reference [40] in the subsequent study.

### 5. Conclusions

(1) By comparing the correlation coefficients of the lettuce canopy with the background (pot, soil, ground) spectral reflectance curves, correlation coefficients differences were used to segment the background region. The mean value for the area overlap measure (AOM) was 0.9254 with a variance of 0.0275, while the mean value of misclassified error (ME) was 0.0292 with a variance of 0.0143. Segmentation accuracy and stability improved compared to traditional reflectance thresholding segmentation methods for hyperspectral images of lettuce canopies collected under outdoor conditions.

(2) The 1st derivative method can better suppress stray light and baseline drift in the spectral data, and is suitable for outdoor hyperspectral data preprocessing. After preprocessing of the 1st derivatives and then correlating the variability of the spectra with the variability of the attributes to be measured by mean-centering, the modeling set $R_C$ of the canopy water content model is 82.21% and the prediction set $R_P$ is 81.25%. The modeling set $R_C$ and prediction set $R_P$ improved by 7.773% and 2.407%, respectively, compared to the canopy water content model constructed from the original data.

(3) The PLS model constructed with the feature wavelengths selected by MCUVE–CARS–PLS had the best comprehensive performance, since this model extracted fewer feature wavelength variables, and had an improved and balanced prediction effect for both the prediction and the modeling sets. The prediction set model correlation coefficient $R_P$ was 84.29% and the root mean square error RMSEP was 0.8627. The modeling set correlation coefficient $R_C$ was 82.71% and the root mean square error RMSECV was 0.7049. The MCUVE–CARS–PLS model was used to calculate the dry basis water content of each pixel point of the lettuce canopy for visual prediction. This study has important practical applications, such as providing a reference for spectral image processing techniques, specifically for lettuce canopy water content prediction under complex outdoor lighting conditions.

**Author Contributions:** Conceptualization, J.Z. and H.L.; methodology, J.Z., C.C., Y.P. and X.Z.; validation, Y.P.; formal analysis, J.Z., C.C. and Y.P.; investigation, J.Z., C.C. and Y.P.; data curation, J.Z.; writing—original draft preparation, J.Z.; writing—review and editing, C.C.; visualization, J.Z.; supervision, C.C.; project administration, H.L. All authors have read and agreed to the published version of the manuscript.

**Funding:** This work was supported by the National Natural Science Foundation of China (grant number 51939005); the Key Research and Development Program of Jiangsu Province (grant number BE2021340, BE2021379); and Demonstration and Promotion Project of Modern Agricultural Machinery, Equipment, and Technology of Jiangsu Province (NJ2021-24).

**Institutional Review Board Statement:** Not applicable.

**Data Availability Statement:** Not applicable.

**Conflicts of Interest:** The authors declare no conflict of interest.

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
