# Peer review of "Detection of Water Content in Lettuce Canopies Based on Hyperspectral Imaging Technology under Outdoor Conditions"

_agriculture, doi:10.3390/agriculture12111796_

Round 1
Reviewer 1 Report
agriculture-1943050-peer-review-v1
The authors present a study to determine water content in lettuce under controlled open field conditions. The study has an interesting idea but lacks the justification in the introduction, the discussion and conclusion sections are not added. The manuscript needs to be revised then resubmitted.
Abstract
Values of R need to be reformatted to % as this is less confusing for readers.
Introduction
More information is needed about previous studies to assess leafy vegetables especially lettuce.
The authors need to justify the need or gap in the research or industry by which their study covers.
Materials and Methods
Explain the meaning of SLD109 and SLD012
A figure of planting stages will help readers follow the germination and growth stages.
In the hardware part, state: model (company, city, country)
Was the same plant imaged many times during growth? If this is the case how did you manage to measure the moisture content? This needs to be fully explained.
Add a figure for the extraction of different objects in the image, i.e. plant, pot, soil, ground, etc.,
Preprocessing methods can be combined in one section
Results
The authors did a good job stating the results in the study.
However, the quality of the figures need to be enhanced as the font is still small. The opposite case is in the tables. This needs to be fixed in all the manuscript.
Discussion
No studies to compare the results with? In any leafy vegetables? This is not appropriate.
Conclusion
The conclusion section is not present in the manuscript.
Reviewer 2 Report
In the article "Detection of water content in lettuce canopies based on hyper-spectral imaging technology under outdoor conditions" (agriculture-1943050) the authors proposed a method to predict lettuce canopy water content by collecting outdoor hyperspectral images of potted lettuce plants and combining spectral analysis techniques and model training methods.
The article has a good introduction and the experimental setup was well designed and the data obtained support the study and the conclusions.
However, there are some issues:
- However, it would have been an added value to use other methods presented in the introduction in the same experimental setup in order to compare and strengthen the claim that the methodology presented is better than the others in practical outdoor applications.
- In section 2.6 it is stated that "... a large number of data variables and overlapping spectral information, and such redundant data must be removed..." but only 4 samples were excluded. Does excluding these samples (4 out of 139) improve the process? Is the computational effort to eliminate them worthwhile?
- Better describe (for example indicating the manufacturer) the elements used (SLD109+SLD012, HM-WSY, etc.).
- When describing the variables used in the equations, it would be better to start the sentence (after the equation) with "where, ..." instead of always using "In the formula...". In equation (1) put that description in a sentence.
- A paragraph describing the next section (or sections) should be placed between a section and a subsection (e.g. between 3. and 3.1, between 3.2 and 3.2.1, etc.).
Round 2
Reviewer 1 Report
In general, the authors addressed most of the comments stated before. However, some deficiencies are still needed to be resolved before the manuscript can be accepted for publication:
1- Correlation coefficient stated in the manuscript is between predicted and measured values which is unlikely to be a –ve. Again, you need to format the r values to % especially with 4 decimal numbers.
2- The discussion section is still not enough. You have to state more studies in the literature to compare you results with. You can use other techniques used to measure water content in lettuce, or other leafy vegetables even by using other techniques than hyperspectral imaging
Author Response
请参阅附件。
